# Identification of Copy Number Variations in Four Horse Breed Populations in South Korea

**DOI:** 10.3390/ani12243501

**Published:** 2022-12-12

**Authors:** Yong-Min Kim, Seok-Joo Ha, Ha-Seung Seong, Jae-Young Choi, Hee-Jung Baek, Byoung-Chul Yang, Jung-Woo Choi, Nam-Young Kim

**Affiliations:** 1Swine Science Division, National Institute of Animal Science, Rural Development Administration, Cheonan 31000, Republic of Korea; 2Department of Animal Science, College of Animal Life Sciences, Kangwon National University, Chuncheon 24341, Republic of Korea; 3Subtropical Livestock Research Institute, National Institute of Animal Science, RDA, Jeju 63242, Republic of Korea

**Keywords:** CNV, structural variation, Jeju horse, thoroughbred, crossbred

## Abstract

**Simple Summary:**

The objective of this study is to detect copy number variations (CNVs) in four horse populations (Jeju horses, Thoroughbreds, Jeju riding horses, and Hanla horses) in South Korea. We found a total of 843 CNV regions (CNVRs) (164.3 Mb), which coincided with 7.2% of the reference horse genome. Overall, copy number losses were found more than gains and mixed CNVRs. A comparison of the CNVRs among the populations showed that a substantial number of CNVRs overlapped each other, while some CNVRs were found specifically in each population. We retrieved parts of CNVRs that overlapped with genes; these overlapping areas are potentially associated with traits of interest in horses. The Thoroughbred and crossbred populations had shared CNVRs overlapping with QTLs (Quantitative trait loci) that were associated with withers height and racing performance. Using gene ontology (GO) analysis, a total of 1884 functional genes were identified within the 577 CNVRs. GO analysis further showed that several of the genes are involved in the olfactory pathway and the nervous system.

**Abstract:**

In this study, genome-wide CNVs were identified using a total of 469 horses from four horse populations (Jeju horses, Thoroughbreds, Jeju riding horses, and Hanla horses). We detected a total of 843 CNVRs throughout all autosomes: 281, 30, 301, and 310 CNVRs for Jeju horses, Thoroughbreds, Jeju riding horses, and Hanla horses, respectively. Of the total CNVRs, copy number losses were found to be the most abundant (48.99%), while gains and mixed CNVRs accounted for 41.04% and 9.96% of the total CNVRs, respectively. The length of the CNVRs ranged from 0.39 kb to 2.8 Mb, while approximately 7.2% of the reference horse genome assembly was covered by the total CNVRs. By comparing the CNVRs among the populations, we found a significant portion of the CNVRs (30.13%) overlapped; the highest number of shared CNVRs was between Hanla horses and Jeju riding horses. When compared with the horse CNVRs of previous studies, 26.8% of CNVRs were found to be uniquely detected in this study. The CNVRs were not randomly distributed throughout the genome; in particular, the *Equus caballus* autosome (ECA) 7 comprised the largest proportion of its genome (16.3%), while ECA 24 comprised the smallest (0.7%). Furthermore, functional analysis was applied to CNVRs that overlapped with genes (genic-CNVRs); these overlapping areas may be potentially associated with the olfactory pathway and nervous system. A racing performance QTL was detected in a CNVR of Thoroughbreds, Jeju riding horses, and Hanla horses, and the CNVR value was mixed for three breeds.

## 1. Introduction

There is abundant evidence that phenotypic diversity can be in part attributed to genetic variation such as single nucleotide polymorphisms (SNPs) and copy number variations (CNVs) [1,2,3]. CNVs have typically been defined as DNA segments larger than 1 kb in length that have differing numbers of copies among individuals of a species. Recently, this definition has been modified to obtain a higher resolution (50 bp) via recent advances in high-density chip arrays and massively parallel sequencing technologies [4,5].

With the release of an equine reference genome assembly and subsequent massive SNP detection, various high-throughput SNP genotyping arrays were developed, enabling rapid genotyping at a reasonable price for diverse horse breeds [6,7,8]. Subsequently, this has led to much research aimed at locating genes associated with horse performance traits, including speed index and durability, via genome-wide association studies [9,10]. However, there are still very limited numbers of publications available for horse CNVs, while they have been intensely investigated in human genomes.

In horse genomics, there are currently several published CNV studies that have used various technologies. Doan et al. (2012) published the first study to detect genome-wide CNVs in horses, using a whole-exome tiling array and an array-comparative genomic hybridization platform. Since the development of high-density SNP arrays for horses [7,11], CNVs could be also assessed throughout the genomes of various horse breeds, of which some studies investigated potential associations with traits of interest in horses, including body size and genetic risk factors for insect-bite hypersensitivity [12,13,14,15].

The Jeju horse (also known as the Jeju pony) was registered with the Food and Agricultural Organization of the United Nations as the only indigenous horse breed in South Korea. It is a small- to medium-sized breed (mean withers height: 116 cm) with a range of colors. The origin of the current Jeju horse remains to be further clarified, though it is known to have involved Mongolian horses [16,17]. Because of a remarkably reduced population, this breed was designated as a Korean natural monument (No. 347) in 1986. In an effort to utilize this native genetic resource in South Korea, the Jeju horse was crossed with Thoroughbreds. A racecourse was opened for horseback racing of Jeju horses and Jeju crossbreeds in 1990, and this led to an increase in the crossbreed population on Jeju Island. However, to our current knowledge, there is still no statutory regulation to clarify exact breed compositions of the crossbreeds, which are usually categorized as either riding horses for outdoor leisure or as racing horses, with respect to their genetic make-up (usually judged by how much Thoroughbred genetics contribute to the crossbreed). It has been reported that most of the crossbreeds used for horseback racing (usually referred to as Hanla horses) on Jeju Island are genetically more heavily influenced by Thoroughbreds than Jeju horses [18]; this is not so surprising, because Thoroughbreds are the most prominent breed for horse racing because of their agility and speed. Conversely, the crossbreed mostly used in riding, which tends to be smaller than the Hanla, is generally called the Jeju riding horse.

The main purpose of this study was to identify CNVs and investigate their patterns throughout the genome of four horse populations, including the Jeju, Thoroughbred, Jeju riding horse, and Hanla populations in the Republic of Korea. We used a total of 469 horse genotyping datasets derived from the Illumina GGP EquineSNP70 Genotyping BeadChip array. Furthermore, functional enrichment analysis for CNVRs that are potentially associated with genes was performed.

## 2. Materials and Methods

### 2.1. Samples and Genotyping

We used a total of 469 individual genotypic datasets using the Illumina GGP Equine SNP70 Bead-chip array (Illumina, San Diego, CA, USA); all of the animals were obtained from the Subtropical Livestock Research Institute in the National Institute of Animal Science in the Republic of Korea. Of the total datasets, we newly genotyped in this study a total of 130 individuals that included Jeju horse (*n* = 4), Thoroughbred (*n* = 5), Jeju riding horse (*n* = 119), and Hanla horse (*n* = 2). We extracted genomic DNA from blood samples treated with EDTA. The samples’ DNA concentration was adjusted to 50 ng/L. Furthermore, Thoroughbred (*n* = 134), Jeju riding horse (*n* = 63), and Hanla horse (*n* = 142) individual genotype data were included from a previous study [18]. Illumina GGP Equine SNP70 Beadchip, which includes approximately 65 k SNPs, was used for genotyping, and Illumina GenomeStudio software v.2.0 (Illumina, San Diego, CA, USA) was used to determine SNP genotypes.

### 2.2. Analysis of Population Genetic Structure

Principal Component Analysis (PCA) and admixture analysis were used to analyze the demographic structure of the horse populations on Jeju Island. The ggplot function in the R package [19] was used to show the relationships between the PC1 and PC2 coordinates after performing the “--pca” flag using PLINK v.1.9 [20] to produce eigenvectors and eigenvalues.

Furthermore, supervised ADMIXTURE (version 1.3) [21] was used to compute the proportions of ancestry (K) at K = 2 for crossbred populations (Hanla horse and Jeju riding horse) to confirm their admixed status. The result derived from ADMIXTURE was visualized using R plots.

### 2.3. CNV Detection

PennCNV v.1.0.5 (https://penncnv.openbioinformatics.org/en/latest/ (accessed on 1 August 2022)) [22], which is based on the Hidden Markov Model (HMM), was used to perform CNV calling. Four input files were used for running the PennCNV software: the signal intensity file (including the SNP Name, log R ratio (LRR), and B allele frequency (BAF)), the population frequency of the B allele (PFB) file, the SNP MAP file (includes the SNP Name, Chromosome, and Position) and the GC-Content file. SNP name, chromosomal position, BAF, and LRR data files were exported as a single file and divided using the PennCNV package’s split option. The PFB file was created using the PennCNV “compile_pfb.pl” function and the GC-Content file was determined as the proportion of GC content both 1 MB upstream and downstream of the SNP locus in the reference genome (UCSC Genome Browser Downloads: https://hgdownload.soe.ucsc.edu/goldenPath/equCab3/bigZips/ (accessed on 30 July 2022). PennCNV was run according to the default criteria using the command line “detect_cnv.pl”. Quality control was performed with “filter_cnv.pl” functions on the samples with a standard deviation of the Log Ratio (SD LRR) >0.3, BAF drift >0.01, and a wave factor (WF) >0.05. Further filtering was performed and CNVs with consecutive SNPs ≥3 and CNV length ≥10 kb were retained. Then “clean_cnv.pl” script was used to combine the adjacent CNVs that had a gap size of less than 20% of the total length of the CNVs. As a result, a total of 45 individuals were filtered (Thoroughbred (*n* = 19), Jeju riding horse (*n* = 25), Hanla (*n* = 1)), and 424 horses were included in the dataset.

### 2.4. CNVR Detection

After CNV detection, the CNVRs were detected by running HandyCNV [23]. “call_cnvr” generates CNV regions as the union of sets of CNVs that overlap by at least one base pair [1]. Each CNVR should have the same boundaries so that each individual can be classified as a diploid, CNVR-gain position (duplications), or CNVR-loss position (deletions). Gain and loss CNVRs that overlapped were combined into a single region to account for genomic regions where both events had happened (CNVR “mixed”). 

CNVRs have shared regions among each breed. The overlapped CNVRs were identified using BEDtools v2.17.0 (Quinlan laboratory, Salt Lake, UT, USA) [24]. We created one comprehensive CNVR table for further analysis. We calculated the sum of all CNVRs in a detected chromosome to determine the genomic percentage covered by CNVRs at the chromosomal level. The gene content of the CNVRs was evaluated using the reference genome EquCab 3.0 from the UCSC Genome Browser Gateway (UCSC Genomics Institute University of California, Santa Cruz, CA, USA).

### 2.5. Gene Contents and Functional Annotation in CNVRs

The NCBI GFF file was used to execute gene annotation to find genes that overlapped the detected CNVRs (https://www.ncbi.nlm.nih.gov/assembly/GCF_002863925.1/genome_assemblies_genome_gff (accessed on 30 July 2022)). Furthermore, the horse quantitative trait locus (QTL) information was annotated using the animal QTL database (https://www.animalgenome.org/cgi-bin/QTLdb/EC/index (accessed on 20 October 2022), on EC_3.0 database). PANTHER17.0 (Protein Analysis THrough Evolutionary Relationships, version 17.0, http://www.pantherdb.org/ (accessed on 20 October 2022)) provided the functional enrichment and gene functional annotation of the CNVRs. Therefore, we conducted gene ontology (GO) analyses for the genes in CNVRs to offer insight into the functional enrichment of the CNVRs [25]. The threshold was set as false discovery rate (FDR) corrected *p*-value < 0.05 and genes were classified by their biological process, molecular function, and cellular component.

## 3. Results and Discussion

### 3.1. Breed Compositions Used in this Study

In this study, we used four horse populations inhabiting Jeju Island. As aforementioned, the Jeju riding horse and Hanla are the crossbreeds of Thoroughbreds and Jeju horses, despite their different main uses (riding and racing for those breeds, respectively) in the Korean horse industry. Namely, the Hanla horse is considered an independent breed in various previous studies [26,27]. In this study, it was reconfirmed via PCA and ADMIXTURE analyses that the Hanla horse forms an independent cluster (Appendix A). However, in the case of the Jeju riding horse, it showed that the pattern was somewhat more irregular than that of the Hanla horse; this pattern was similarly observed in the ADMIXTURE analysis (Appendix A). These results are not surprising, because there is no accurate documentation to enable such a distinction, particularly for the Jeju riding horse. Although it seems obvious that the Jeju riding horse has been less influenced by Thoroughbred genetics than the Hanla, it is difficult to clearly define the mixing ratio of Thoroughbred and Jeju horse. Therefore, further analysis is required to suggest a clear guideline to designate breed composition, particularly for the Jeju riding horse population.

### 3.2. Identifying Genome-Wide CNVs

CNVs are structural variations that are a primary source of genetic diversity and phenotypic variation. As such, CNVs are recognized as significant for identifying genetic diversity among populations and in the evolution of breeds [28,29,30]. To profile genome-wide CNVs, we first detected CNVs throughout 31 autosomes in each of the four horse populations on Jeju Island, including Jeju horses, Thoroughbreds, Jeju riding horses, and Hanla horses, using PennCNV v1.0.5 [22]. A total of 4482 CNVs were detected for the four horse breed populations; we found that copy number gains (2805 CNVs) were more abundant than losses (1677 CNVs). The higher frequency of gains compared with losses was observed for most of the horse populations except Thoroughbreds (Appendix A).

These results are similar to previous studies showing a high incidence of gains in horse populations. It has been suggested that gains in coding or enhancing sequences increase the genetic diversity of organisms, resulting in phenotypic variety and the putative potential to adapt in challenging environments [31]. For this reason, Jeju riding horses and Hanla horses are considered to be undergoing the process of removing mutations and adapting to the environment by artificial selection. 

Of the four breeds, the Hanla showed the highest number of CNVs (1818 CNVs with 493 losses and 1325 gains) and Jeju horses had the lowest (41 CNVs with 11 losses and 30 gains). The difference in CNVs indicates that genetic variation from CNVs may contribute to breed phenotypic diversity, but it may also result from the different demographic history and effective population sizes between breeds [32]. The very small number of CNVs observed in this study for Jeju horses may be a result of the small number of samples. The CNVs ranged in size from 11 kb in Thoroughbreds to 2816 kb in Jeju riding horses. The total length of CNVs ranged from 12 Mb in Jeju horses to 376 Mb in Hanla horses, which is consistent with the range of genome coverage from 0.51% to 16.46% for Jeju horses and Hanla horses, respectively (Table 1).

### 3.3. Defining CNVRs

To detect CNVRs for each horse population, we merged CNVs that overlapped by at least one base pair. A total of 992 CNVRs were detected from all four populations. Hanla horses showed the highest number of CNVRs (310 CNVRs with 170 gains, 127 losses, and 13 mixed), and Jeju horses had the lowest number of CNVRs (30 CNVRs with 19 gains and 11 losses). The CNVR coverage of chromosomes is 0.24% (5.48 Mb) in Jeju horses, 3.2% (73.53 Mb) in Thoroughbreds, 3.47% (97.11 Mb) in Jeju riding horses, and 2.91% (66.32 Mb) in Hanla horses. Additionally, the longest CNVRs were identified in Jeju riding horses (2866 kb) and Hanla horses (2569 kb), while Thoroughbreds and Jeju horses had the least (2362 kb and 2156 kb, respectively) (Table 2). The CNVR distributions appear to be affected by the sample size; note that the small number of Jeju horses affected the very small number of CNVs detected and the CNVR distribution in the Jeju horse population.

In each chromosome, the number of CNVRs was the highest on ECA1 in Thoroughbreds (*n* = 25), Jeju horses (*n* = 5), and Hanla horses (*n* = 34), and the number of CNVRs was the highest on ECA4 in Jeju riding horses. Conversely, the coverage of CNVRs on each chromosome was the highest on ECA12 (5.8% of the Jeju horse genome, 9.0% of the Jeju riding horse genome, and 8.4% of the Hanla horse genome) and on ECA7 (10.3% of the Thoroughbred genome) (Appendix A, Figure 1).

As observed in prior studies with horse breeds, the largest shared CNVRs were discovered on ECA12 [33,34]. ECA12 displays the particular characteristic of being enriched with clusters of olfactory receptor genes, which is also observed in other mammalian genomes, and it has been proposed that this property affects the fight or flight response and temperament diversity in horses [35]. Therefore, the mild personality of Jeju horses, could be the result of selection for riding in a wide variety of environments.

We assessed CNVRs that overlapped among the four populations (overlapping CNVRs). A total of 843 overlapping CNVRs was retrieved by considering CNVRs that overlapped by at least one base pair between populations. The overlapping CNVRs comprised 346 gains, 413 losses, and 84 mixed events (a mean length of 195 kb), and ranged from 0.4 kb to 2866 kb in size and covered 164,330 kb of the horse genome, which corresponds to 7.2% of the horse autosomes. 

In each breed, CNVs and CNVRs showed more gain in the horse population (Table 1 and Table 2). However, 843 overlapping CNVRs show more loss. Thoroughbred populations have long loss-CNVRs (more than 0.3 Mb, *n* = 60). Therefore, among overlapping CNVRs, there was an increase in loss-CNVRs (Figure 2).

The distribution of CNVRs across all autosomes varied considerably, with the highest number at 74 on ECA4 and the lowest at 4 on ECA24. The ratio of total estimated CNVR length per chromosome to the length of that chromosome varied from 16.3% for ECA7 to 0.7% for ECA24 (Figure 1). 

Among the 843 CNVRs, 589 (70.6%) were specific CNVRs that did not overlap with other breeds and appeared only in each breed as follows: Jeju horses (*n* = 5), which had the fewest specific CNVRs, Thoroughbreds (*n* = 180), Jeju riding horses (*n* = 202), and Hanla horses (*n* = 202) (Figure 3, Appendix A). Because Jeju riding horses and Hanla horses are crossbreeds produced by hybridizing Jeju horses and Thoroughbreds, we can expect these two breeds to exhibit significant characteristics in CNVRs. The majority of horse breeds were of recent origin and had undergone significant crossbreeding before closed breeds were founded, which has resulted in a high level of haplotype sharing [6,36], and thus fewer breed-specific CNVRs can be found than crossbred [37].

The breed-specific CNVR for each breed showed high losses for Jeju horses (80%), Thoroughbreds (82.78%), and Hanla horses (50.99%), while Jeju riding horses showed a gain of 49.01%. We confirmed that the CNVRs shared by the populations had a gain of 48.82%, confirming that CNVR gains are shared more among these populations than losses. The results of the present study were in agreement with the findings of Wang et al. (2014) [38] and Ghose et al. (2014) [34], who reported that losses prevailed over gains in most horse breeds.

When comparing CNVRs shared between breeds, the smallest number was found between Thoroughbreds and Jeju horses (16 CNVRs), and the largest number was found between Hanla and Jeju riding horses (168 CNVRs). Thoroughbreds shared 130 CNVRs with Jeju riding horses and 128 CNVRs with Hanla horses. In contrast, compared with Jeju horses, a relatively small number (22 CNVRs) were shared between Jeju horses and Jeju riding horses, and 21 CNVRs were shared between Jeju horses and Hanla horses (Figure 3). Jeju horses and Thoroughbreds are parental breeds, and they showed the lowest number of shared CNVRs because of their low genetic association. Conversely, Hanla and Jeju riding horses shared more CNVRs with Thoroughbreds than with Jeju horses. We conclude that this is because these breeds were produced with a heavier influence from Thoroughbreds than Jeju horses, with the aim of breeding horses for riding and racing purposes.

### 3.4. Comparison of CNVRs to those Identified in Previous Studies

To characterize the CNVRs identified in this study in more detail, we compared them to those identified in ten previous studies that used various methods [13,15,33,34,35,38,39,40,41,42]. A total of 2519 CNVRs were identified across all 11 studies: 1844 (73.2%) CNVRs were identified in our study and the previous studies, and 675 (26.8%) CNVRs were identified only in this study (Appendix A). The number of CNVRs found in this study was higher than that detected by Kader et al. (2016) (122 CNVRs) [41]. The number of CNVRs was smaller than that detected by Schurink et al. (2018) (5350 CNVRs), while the genome coverage rate of CNVRs was similar to Schurink et al.’s (2018) study [13].

When our results were compared individually with the results of the previous studies, the highest matching rate (approximately 33.99%) was with Solé et al. (2019). The lowest matching rate (approximately 4.22%) was with Wang et al.’s (2014) study (Table 3).

These matching rates may be the result of breed differences or the genome coverage ratio of the CNVRs in the various studies. Solé et al. (2019) [33] had the largest coverage ratio, which may have contributed to the high matching rate with our study; conversely, Wang et al. (2014) [38] had the smallest coverage ratio. With only a few exceptions, the higher the coverage, the higher the matching rate.

Comparing our CNVRs with other studies, we could confirm the CNVRs that overlapped with previous studies as well as discovered only in this study. The non-overlapped CNVRs with previous studies had unique genetic features of the Korean horse populations used in this study. However, we consider that this is not an accurate reflection of actual breed differences, because there are differences (such as sample sizes, platform, and filtering criteria) in the conditions for detecting CNVRs between each study. Therefore, it is necessary to only compare studies that have the same conditions, such as platform and CNV and CNVR analysis algorithms.

### 3.5. QTLs Overlapping with CNVRs

We annotated previously reported horse QTLs to a total of 843 CNVRs, which were discovered throughout 31 autosomes of the four horse populations in this study (Appendix A). We retrieved QTLs that overlapped with CNVRs, and we found several CNVRs that might be associated with phenotypic or economically desirable traits in horses. Because of the inheritance of a large proportion of CNVs, we have focused on CNVRs in two groups to observe CNVRs that might be derived from parental breeds (Jeju horses and Thoroughbreds): one with CNVRs shared between the Thoroughbred and crossbred populations (Jeju riding horses, Hanla horses, or both) and the other with CNVRs shared between the Jeju horse and crossbred populations.

Among the CNVRs shared between the Thoroughbred and crossbred populations (Jeju riding horses and Hanla horses), we identified 23 CNVRs (gains or losses) overlapping with QTLs that were previously reported to be associated with withers height. These regions were located on ECA7 (50.8–51.4 Mb; 73.5–73.6 Mb), ECA8 (51.5–51.6 Mb), ECA16 (34.1–34.9 Mb), and ECA26 (38.5–38.6 Mb). We also found one mixed-type CNVR (ECA20: 32.7–33.0 Mb) that overlapped with the QTL associated with racing performance. As the Thoroughbred is one of the main representative racing horse breeds, this CNVR in crossbred populations might be influenced by Thoroughbreds. For CNVRs shared between the Jeju horse and Jeju riding horse populations, we located one loss-type CNVR (ECA3: 38.8–39.0 Mb) shared between Jeju horses and Jeju riding horses that was associated with a QTL reported to have an association with white coat markings (Appendix A).

### 3.6. Functional Annotation for CNVRs

Based on the NCBI (National Center for Biotechnology Information) annotation of the EquCab 3.0 genome, 1884 genes overlapped with 843 CNVRs (68.48%). We conducted functional annotation analysis on 1,884 genes using PANTHER. As a result, the most significantly enriched biological processes were included in three main categories: sensory perception of smell (raw *p*-value: 3.93 × 10^−5^) and chemical stimulus (raw *p*-value: 4.91 × 10^−5^), and nervous system processes (raw *p*-value: 4.42 × 10^−5^) (Table 4).

Furthermore, when analyzing the function of genes present in CNVRs for each breed, GO with the highest enrichment in Jeju horses had many functions related to G-protein functions, such as “G-protein-coupled serotonin receptor signaling pathway”, in the biological process. Furthermore, when analyzing the function of genes present in the CNVRs for each breed, Thoroughbreds had many functions related to olfactory senses such as “detection of chemical stimulus invalidated in sensory perception of smell” and “sense perception of smell” in the biological process. The Jeju riding horses had immunological and olfactory functions such as “antimicrobial humoral immune response mediated by antimicrobial peptide” and “detection of chemical stimulation in sensory perception of smell”. Additionally, we identified olfactory-related functions in Hanla horses, such as “sense perception of smell, detection of chemical stimuli invalidated in sense perception of smell” (Appendix A). Previous studies have shown that olfactory receptor and immune-related genes were located in CNVRs, and the results of our ontology study support this. In this study, additional genes related to the nervous system were annotated. In Jeju horses, we confirmed that G-protein-related genes were identified more than olfactory and immune functions, and in contrast, all varieties except Jeju horses showed a strong correlation with smell-related functions.

## 4. Conclusions

The characteristics of CNVs and CNV regions in the Korean horse populations of Jeju (Jeju horse, Thoroughbred, Jeju riding horse, and Hanla horse) were investigated using the Illumina GGP Equine SNP70 Beadchip in this study. We detected a total of 843 CNVRs throughout 31 autosomes, and these CNVRs covered approximately 7.2% of the Equine genome. The discovered CNVRs include both previously reported and novel CNVRs. These CNVRs overlapped with known horse QTLs such as those for withers height, racing performance, and white coat markings, and functional analyses revealed that genes associated with olfactory function and nervous response were highly expressed in CNVRs. The results derived from this study will extend understanding of the genetic composition of Korean horse populations and unique CNVRs throughout the horse genome; furthermore, it will provide resources for future studies on CNVs in horses.

## Figures and Tables

**Figure 1 animals-12-03501-f001:**
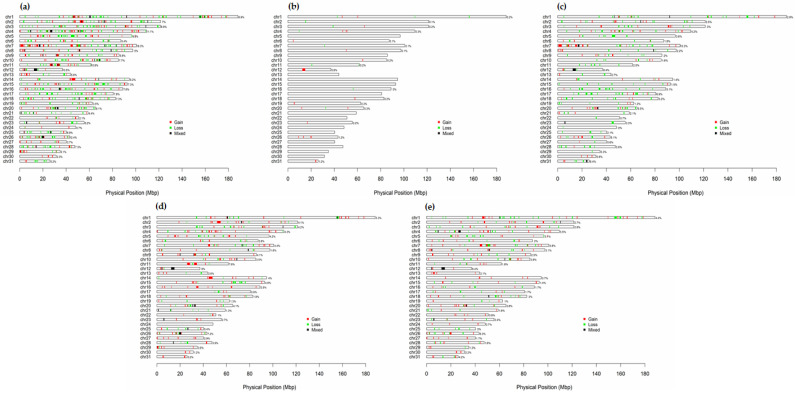
Plot of copy number variation regions (CNVR) on 31 horse (*Equus caballus*) autosomes. (**a**) Total CNVR distribution; (**b**) Jeju horse CNVR distribution; (**c**) Thoroughbred CNVR distribution; (**d**) Jeju riding horse CNVR distribution; (**e**) Hanla horse CNVR distribution. Red, green, and black represent gains, losses, and mixed CNVRs, respectively.

**Figure 2 animals-12-03501-f002:**
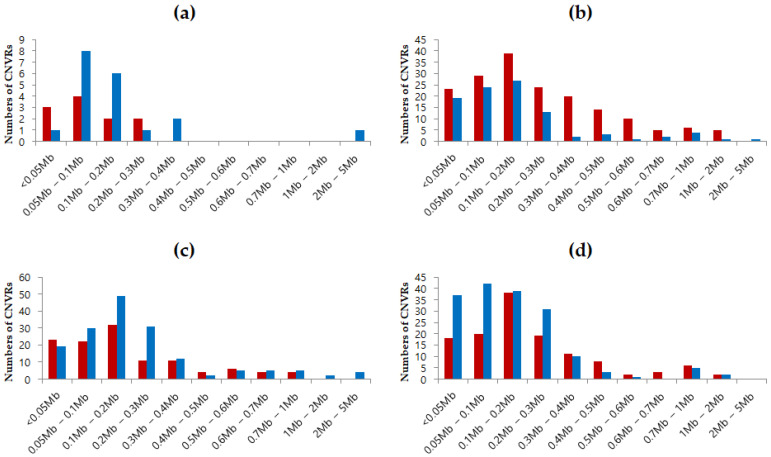
Gain and loss distribution by length of CNV region (CNVR) in four horse breeds in South Korea. (**a**) Jeju horse CNVR distribution; (**b**) Thoroughbred CNVR distribution; (**c**) Jeju riding horse CNVR distribution; (**d**) Hanla horse CNVR distribution. Red and blue represent losses and gains, respectively.

**Figure 3 animals-12-03501-f003:**
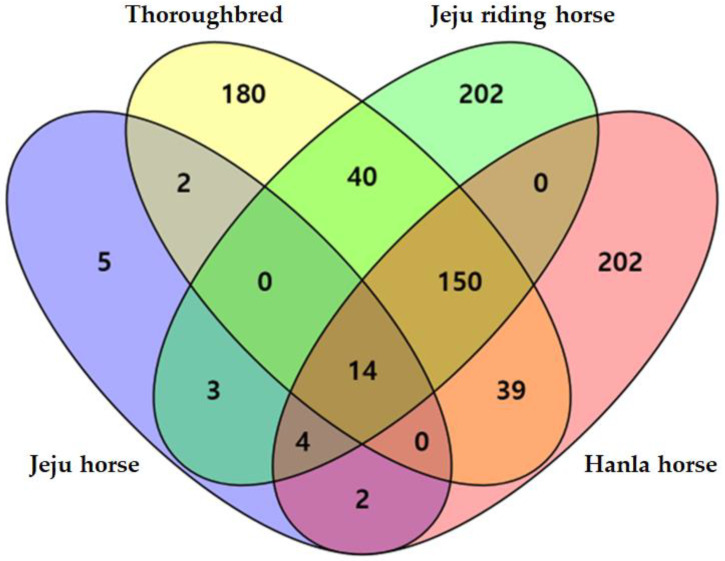
Number of overlapping copy number variation regions (CNVRs) between four breeds in South Korea. Purple is number of Jeju horse CNVR; Yellow is number of Thoroughbred CNVR; Green is number of Jeju riding horse CNVR; Red is number of Hanla horse CNVR, respectively.

**Table 1 animals-12-03501-t001:** Copy number variation (CNV) analysis of four horse breeds in South Korea.

Breed	Sample	CNV	CNL ^1^	CNG ^2^	Length Min ^3^	Length Max ^4^	Length Median ^5^	Total Length in CNV	Length Average
Jeju horse	4	41	11	30	12,219	2,156,704	116,562	11,529,371	281,204.17
Thoroughbred	120	1340	716	624	10,509	2,563,439	155,989	374,113,603	279,189.26
Jeju riding horse	157	1283	457	826	10,718	2,816,442	147,819	358,366,183	279,318.93
Hanla horse	143	1818	493	1325	10,509	2,246,935	108,623	375,503,162	206,547.39

^1^ CNL, CNV losses. ^2^ CNG, CNV gains. ^3^ Length Min, Minimum length of CNV. ^4^ Length Max, Maximum length of CNV. ^5^ Length Median, Median length of CNV.

**Table 2 animals-12-03501-t002:** The basic statistic of copy number variation regions (CNVRs) of four horse breeds in South Korea.

Breed	CNVR	Gain	Loss	Mixed	Length Min ^1^	Length Max ^2^	Length Median ^3^	Total Length in CNVR	CNVR Coverage ^4^
Jeju horse	30	19	11	0	12,219	2,156,704	94,014	5,475,931	0.24%
Thoroughbred	281	97	175	9	10,509	2,563,439	163,969	73,526,600	3.22%
Jeju riding horse	301	164	117	20	10,718	2,865,712	161,474	79,107,681	3.47%
Hanla horse	310	170	127	13	10,509	2,569,249	134,033	66,321,742	2.91%

^1^ Length Min: Minimum length of CNVR. ^2^ Length Max: Maximum length of CNVR. ^3^ Length Median: Median length of CNVR. ^4^ CNVR frequency in horse genome (Horse genome length/Total length in CNVR)

**Table 3 animals-12-03501-t003:** Comparison of reported copy number variation regions (CNVRs) for horses (*Equus caballus*).

Study	Platform	Breed	Sample	CNV Analysis Algorithm	CNVR Count	CNVR Range (kb~Mb)	Genome Coverage %	Reference Genome	Overlapped CNVR Count with the Present Study	Overlapped CNVR Percentage with the Present Study
Doan et al. (2012)	Array CGH	15	16	ADM-2	775	0.2–3.5	3.7	EquCab 2.0	169	9.53%
Dupuis et al. (2013)	Illumina Equine 50 K SNP BeadChip	4	477	PennCNV	478	0.1–2.7	2.3	EquCab 2.0	160	8.90%
Ghosh et al. (2014)	Array CGH	16	38	Agilent Genomic Workbench	258	1–2.5	1.2	EquCab 2.0	97	7.44%
Wang et al. (2014)	Array CGH	6	6	segMNT	353	6.1–0.5	0.6	EquCab 2.0	105	4.22%
Kader et al. (2016)	Illumina Equine 70 K SNP BeadChip	3	96	PennCNV	122	0.2–2.2	0.8	EquCab 2.0	76	5.58%
Ghosh et al. (2016)	Array CGH	NA	63	Agilent Genomic Workbench	245	0.1–79.9	6.1	EquCab 2.0	87	7.30%
Schurink et al. (2018)	Axiom Equine Genotyping Array (670,796 SNPs)	1	222	PennCNV	5350	0.12–1.03	11.2	EquCab 2.0	658	18.41%
Solé et al. (2019)	Axiom Equine Genotyping Array (670,796 SNPs)	8	1755	Axiom^®^ CNV summary	939	1–21.3	24.41	EquCab 2.0	360	33.99%
Wang et al. (2022)	Illumina Equine 70 K SNP BeadChip	10	282	PennCNV	495	1–2.3	1.8	EquCab 3.0	206	9.61%
Laceca et al. (2022)	670 K Affymetrix Axiom™ Equine Genotyping Array	1	654	PennCNV	1007	1–4.6	4.4	EquCab 3.0	163	8.87%
Present study	Illumina GGP Equine 70 K SNP BeadChip	4	469	PennCNV	843	0.4–2.9	7.2	EquCab 3.0	-	-

**Table 4 animals-12-03501-t004:** Gene ontology (GO) terms (biological process) for annotated genes in copy number variation regions (CNVRs) of horses.

Category	ID	Term	REFLIST ^1^	Upload ^2^	Expected ^3^	Fold Enrichment ^4^	raw *p*-Value	FDR ^5^
GO_BP	GO:0007608	sensory perception of smell	1039	114	75.35	1.51	3.93 × 10^−5^	5.14 × 10^−2^
GO_BP	GO:0050911	detection of chemical stimulus involved in sensory perception of smell	1021	112	74.04	1.51	4.46 × 10^−5^	4.27 × 10^−2^
GO_BP	GO:0050907	detection of chemical stimulus involved in sensory perception	1063	115	77.09	1.49	6.13 × 10^−5^	4.90 × 10^−2^
GO_BP	GO:0007606	sensory perception of chemical stimulus	1136	122	82.38	1.48	4.91 × 10^−5^	4.41 × 10^−2^
GO_BP	GO:0007600	sensory perception	1407	146	102.03	1.43	4.16 × 10^−5^	4.60 × 10^−2^
GO_BP	GO:0050877	nervous system process	1759	176	127.56	1.38	4.42 × 10^−5^	4.54 × 10^−2^
GO_BP	GO:0051716	cellular response to stimulus	5968	524	432.78	1.21	9.34 × 10^−7^	2.69 × 10^−3^
GO_BP	GO:0007165	signal transduction	4609	404	334.23	1.21	4.92 × 10^−5^	4.16 × 10^−2^
GO_BP	GO:0007154	cell communication	4952	434	359.11	1.21	1.95 × 10^−5^	3.12 × 10^−2^
GO_BP	GO:0023052	signaling	4860	425	352.43	1.21	3.02 × 10^−5^	4.34 × 10^−2^
GO_BP	GO:0032501	multicellular organismal process	5482	472	397.54	1.19	4.11 × 10^−5^	4.93 × 10^−2^
GO_BP	GO:0050896	response to stimulus	7140	613	517.77	1.18	9.50 × 10^−7^	2.28 × 10^−3^
GO_BP	GO:0050794	regulation of cellular process	10,163	833	736.99	1.13	2.09 × 10^−6^	3.75 × 10^−3^
GO_BP	GO:0065007	biological regulation	11,479	930	832.43	1.12	1.06 × 10^−6^	2.18 × 10^−3^

^1^ The number of genes in the reference list that map to this particular annotation data category. ^2^ The number of genes in our uploaded list that map to this annotation data category. ^3^ The column contains the expected value, which is the number of genes expected in our list. ^4^ The column shows the fold enrichment of the genes observed in the uploaded list over the expected value (number in our list divided by the expected number). If it is greater than 1, it indicates that the category is overrepresented. Conversely, the category is underrepresented if it is less than 1. ^5^ The False Discovery Rate (FDR)-corrected values as calculated by the Benjamini-Hochberg procedure.

## Data Availability

The datasets of the current study are available from the corresponding author upon reasonable request and with permission of the National Institute of Animal Science, RDA in the Republic of Korea.

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
