# Peer review of "Identification of Copy Number Variations in Four Horse Breed Populations in South Korea"

_animals, 2022, doi:10.3390/ani12243501_

Round 1

Reviewer 1 Report

The words Copy number variation and horse appear in the title and as keywords. It should only appear in one location.

The graphics in Figure 1 could be a little bigger. The visualization would be better and easier to compare the results.

The distribution of graph B (figure 1) was not affected by the size of the sample? maybe cut it out or enlarge the sample (Jeju horse n=4)?

Conclusion:

The authors did not complete the work. They only show results and report that: "The results derived from this study extend the coverage of CNVRs in the horse genome and provide resources for future studies on CNVs". This needs to be done in a simple and direct way, based on the results obtained. Pondering the title and objectives.

Author Response

Modifying the manuscript, we made changes that were highlighted to address the reviewer’s comments by using blue-color in the manuscript. Below are reviewer 1 comments in italics with bold and our responses in plain text.

Reviewer #1:

- The words Copy number variation and horse appear in the title and as keywords. It should only appear in one location.

We appreciate the reviewer’s comment. As the reviewer pointed out, we have changed the keyword form “Keywords: Copy number variation; Jeju horse; Thoroughbred; Jeju riding horse; Hanla horse” to “Keywords: CNV; Structural variation; Jeju horse; Thoroughbred; Crossbred”

- The graphics in Figure 1 could be a little bigger. The visualization would be better and easier to compare the results.

We thank for the reviewer’s comment. We agree that Figure 1 is somewhat small, which need to be larger. Therefore, we enlarged the Figure 1 for better visualization. Also, we have inserted images for each breed separately to improve such an image quality issue.

- The distribution of graph B (figure 1) was not affected by the size of the sample? maybe cut it out or enlarge the sample (Jeju horse n=4)?

We appreciate the reviewer’s comment. Unfortunately, additional analysis such as modifying or increasing the sample size of Jeju horse for this study is not possible currently.  Although we know that bias may occur due to the number size of Jeju horses, Jeju horses were included in the results because they are an important breed in this study. Therefore, we have added the following sentence after Table 2 in “Results and Discussion” section:

“The CNVR distributions look affected by the sample size; You should be aware of that the small number of Jeju horses affected the very small number of CNVs detected and CNVRs distribution in Jeju horse population.”

- Conclusion: The authors did not complete the work. They only show results and report that: "The results derived from this study extend the coverage of CNVRs in the horse genome and provide resources for future studies on CNVs". This needs to be done in a simple and direct way, based on the results obtained. Pondering the title and objectives.

We do appreciate the reviewer’s comment. As you pointed out, the conclusion was not appropriate enough for the results. Therefore, we have edited those sentences as follows form “The results derived from this study extend the coverage of CNVRs in the horse genome and provide resources for future studies on CNVs.” to “The results derived from this study would extend understanding of genetic composition of Korean horse populations and unique CNVRs throughout the horse genome; furthermore, it would provide resources for future studies on CNVs in horses.”

We hope this revised manuscript will not get the reviewer disappointed.

Reviewer 2 Report

The authors were looking for CNV regions in four horse groups living in South Korea. The reported CNV regions were mostly overlapping but unique regions were also idenified.

CNV regions were also annotated to previously described QTLs, functional annotations are given.

I suggest the MS for publication.

One notice:

I would happily look at the third principal component in Supplementary Figure S1, more exactly PC3 vs. PC1 and PC3 vs. PC2. 

Author Response

Modifying the manuscript, we made changes that were highlighted to address the reviewer’s comments by using blue-color in the manuscript. Below are reviewer 1 comments in italics with bold and our responses in plain text.

Reviewer #2:

- The authors were looking for CNV regions in four horse groups living in South Korea. The reported CNV regions were mostly overlapping but unique regions were also idenified.

CNV regions were also annotated to previously described QTLs, functional annotations are given.

I suggest the MS for publication.

We thank you for the reviewer’s comment. We hope this revised manuscript will not get the reviewer disappointed.

- One notice: I would happily look at the third principal component in Supplementary Figure S1, more exactly PC3 vs. PC1 and PC3 vs. PC2

We appreciate the reviewer’s comment. Following the reviewer’s suggestion, we have added PC3 vs. PC1 and PC3 vs. PC2 results to supplementary figure S1. Therefore, the figure subscription will be changed.

From “Supplementary Figure S1. PCA and admixture analysis results of four horse breeds. (a) PCA analysis; (b) Admixture analysis (K = 2)”

to “Supplementary Figure S1. PCA and admixture analysis results of four horse breeds. (a) PCA analysis of PC1 vs. PC2; (b) PCA analysis of PC3 vs. PC2; (c) PCA analysis of PC3 vs. PC1; (d) Admixture analysis (K = 2)”. They show similar results to PC1 vs. PC2.

Round 2

Reviewer 1 Report

That's ok.

Publish!